# Artificial Fluorescent Glucosinolates (F-GSLs) Are Transported by the Glucosinolate Transporters GTR1/2/3

**DOI:** 10.3390/ijms24020920

**Published:** 2023-01-04

**Authors:** Christa Kanstrup, Claire C. Jimidar, Josip Tomas, Giuliano Cutolo, Christoph Crocoll, Marie Schuler, Philipp Klahn, Arnaud Tatibouët, Hussam Hassan Nour-Eldin

**Affiliations:** 1DynaMo Center, Department of Plant and Environmental Sciences, Faculty of Science, University of Copenhagen, Thorvaldsensvej 40, 1871 Frederiksberg C, Denmark; 2Institute of Organic Chemistry, Technische Universität Carolo Wilhelmina zu Braunschweig, Hagenring 30, 38106 Braunschweig, Germany; 3Institute of Organic and Analytical Chemistry, Université d’Orléans, Rue de Chartres, BP6759, CEDEX 02, 45067 Orléans, France; 4Department of Chemistry and Molecular Biology, Division of Organic and Medicinal Chemistry, University of Gothenburg, Kemigården 4, 412 96 Göteborg, Sweden

**Keywords:** glucosinolate transporters, GTR, fluorescent glucosinolates, fluorescent substrates

## Abstract

The glucosinolate transporters 1/2/3 (GTR1/2/3) from the Nitrate and Peptide transporter Family (NPF) play an essential role in the transport, accumulation, and distribution of the specialized plant metabolite glucosinolates. Due to representing both antinutritional and health-promoting compounds, there is increasing interest in characterizing GTRs from various plant species. We generated seven artificial glucosinolates (either aliphatic or benzenic) bearing different fluorophores (Fluorescein, BODIPY, Rhodamine, Dansylamide, and NBD) and investigated the ability of GTR1/2/3 from *Arabidopsis thaliana* to import the fluorescent glucosinolates (F-GSLs) into oocytes from *Xenopus laevis*. Five out of the seven F-GSLs synthesized were imported by at least one of the GTRs. GTR1 and GTR2 were able to import three F-GSLs actively above external concentration, while GTR3 imported only one actively. Competition assays indicate that the F-GSLs are transported by the same mechanism as non-tagged natural glucosinolates. The GTR-mediated F-GSL uptake is detected via a rapid and sensitive assay only requiring simple fluorescence measurements on a standard plate reader. This is highly useful in investigations of glucosinolate transport function and provides a critical prerequisite for elucidating the relationship between structure and function through high-throughput screening of GTR mutant libraries. The F-GSL themselves may also be suitable for future studies on glucosinolate transport in vivo.

## 1. Introduction

Glucosinolates (GSL) are amino acid-derived, sulfur- and nitrogen-containing thioglucosides with more than 130 identified structures in nature [1]. GSL are found mainly in the order *Brassicales* [2], wherein they function as important defense compounds against pathogens and herbivores [3]. From an agricultural perspective, there is a desire to control the accumulation patterns of glucosinolate in plants. For example, in both established and emerging oilseed crops such as *Brassica napus* (rape), *Brassica juncea* (mustard), and *Camelina Sativa*, the high accumulation of antinutritional glucosinolates in seeds must be eliminated to enable the usage of the otherwise protein-rich seed-meal in animal feed [4,5,6].

GSLs are translocated to the seeds, which are devoid of biosynthesis capability. Accordingly, the identification of the glucosinolate transporters (GTRs), GTR1/AtNPF2.10, GTR2/AtNPF2.11 GTR3/AtNPF2.9 in *Arabidopsis thaliana* [7,8] paved the way for targeting GTRs as a novel strategy for modulating the levels of GSL in *Brassica* crops in a tissue-specific manner [4,9]. To explore this potential, an increasing number of studies are characterizing GTRs from various plant origins [9,10,11,12].

The GTRs were identified via a functional genomics approach that used Liquid Chromatography Mass Spectrometry (LCMS) to detect the uptake of GSLs in *Xenopus laevis* oocytes [7,8,13]. The GTRs are located in the plasma membrane where they import GSL into the cytosol and have been suggested to be involved in phloem loading, leaf distribution, and rhizosecretion of GSL [7,14,15,16]. They belong to the Nitrate and Peptide transporter Family (NPF), where most characterized members are proton-coupled symporters with an electrogen transport process [7,17,18,19,20]. It has been shown that the GTRs co-transport protons together with GSLs, which elicits a net influx of positive charge that can be measured by two-electrode voltage-clamp (TEVC) electrophysiology. Hence, both LCMS and electrophysiology-based transport assays have been used to characterize the transport properties of GTRs from both *Arabidopsis* and other *Brassica* plants. Characterization via LCMS- and electrophysiology-based detection systems is ideal for determining substrate specificity, transport mechanism, and kinetic properties. However, both methods require specialized skills as well as costly equipment and are inherently slow and laborious. In comparison, detecting the activity of transporters of colored- or fluorescent substrates can be performed much faster and cheaper via standard fluorescence plate readers and offers the possibility to perform high-throughput screening of large mutant collections [21,22].

In the context of studies on GSL metabolism, artificial fluorescent GSLs have been synthesized as tools in fluorescence imaging aimed at tracking glucosinolate-metabolizing bacteria in the human gut microbiome. Based on **GSL-N_3_** precursors, different small fluorophores can be attached to the sidechain of different GSL-core structures [23,24]. Notably, the characterization of GTRs has revealed that they are notoriously promiscuous towards the glucosinolates’ amino side chain [7,25]. This prompted us to explore whether artificial fluorescent glucosinolates are accepted as substrates by the GTRs.

Here, we introduce a novel transport assay for detecting GTR activity, which exploits the hitherto unknown ability of the GTRs to transport GSLs that are covalently linked to small fluorophores. We generated seven different fluorescent glucosinolates (F-GSL) using different artificial GSL precursors bearing aliphatic (GSL-A) or benzylic (GSL-B) aglycon moiety and linked them to fluorophores (Figure 1). The level of uptake was investigated for each F-GSL in GTR-expressing oocytes via LCMS and simple fluorescence measurements on a standard plate reader. We show that three of the tested F-GSLs are transported actively at a preference similar to natural GSLs. Notably, we find pairs of F-GSL towards whom GTR1/GTR2 and GTR3 exhibit reciprocal substrate preferences, thus reflecting the natural differences in substrate preference observed for natural GSL.

## 2. Results

### 2.1. Synthesis of Seven Artificial Fluorescent Glucosinolates (F-GSLs)

Different strategies were applied to generate the aliphatic and benzylic fluorescent glucosinolates (F-GSLs). On the first hand, F-GSLs were generated based on simple copper(I)-mediated azide-alkyne click chemistry (CuAAC) linking different alkyne-bearing fluorophores to propyl **GSL-A-N_3_**, an azide containing artificial GSL [23]. Using this strategy, **GSL-A-BODIPY**, **GSL-A-Rhodamine**, and **GSL-A-Dansylamide** were synthesized as outlined in Figure 1, bearing a fluorescently labeled sidechain mimicking an aliphatic ornithine residue. In a different method, the fluorescein fluorophore was condensed on the pentyl **GSL-A-N_3_**, producing the **GSL-A-Fluorescein**. Access to the benzylic GSLs followed similar procedures. We generated **GSL-B-Fluorescein**, **GSL-B-DNS**, and **GSL-B-NBD** from **GSL-B-N_3_** [24] by sequence of *Staudinger* reduction in the presence of triphenylphospine and subsequent reaction of the released **GSL-B-NH_2_** with dansylchloride, NBD chloride, or FITC, as outlined in Figure 2.

In all the F-GSLs synthesized in this study, the fluorophores are attached to the side chains of core structure mimicking two of the more important types of GSLs, namely the L-methionine and L-tyrosine-derived ones with aliphatic and benzylic side chains, respectively. To some extent, the natural GSLs such as Sinalbin (4-Hydroxybenzyl GSL), Glucoaubrietin (4-Methoxybenzyl GSL), or Glucotropaeolin (Benzyl GSL) resemble the **GSL-B** benzylic side chain. On the other hand, Sinigrin (Prop-2-enyl GSL) resembles the **GSL-A** propyl side chain and Glucobrassicanapin (Pent-4-enyl GSL) the **GSL-A** pentyl side chain.

### 2.2. Transport of Fluorescent Glucosinolates Depends on the Fluorophore Attached

Although GTRs are promiscuous towards the side chain [7,25], it is not given that they would accept artificial GSLs to which an entire fluorophore molecule had been added to the side chain. To investigate whether any of the seven F-GSL are transported by the GTRs, we performed transport assays, wherein GTR1, GTR2, and GTR3 expressed in *Xenopus laevis* oocytes were exposed to 50 μM external concentration of each individual F-GSL at pH 5 for 1 h. To make sure the F-GSL are stable and remain intact during the assays, we first conducted three independent experiments where oocyte homogenates were analyzed by LCMS (Figure 2).

For **GSL-A-Rhodamine**, no uptake was detected for any GTR. For **GSL-A-BODIPY**, a relatively high background level was detected in mock oocytes. In comparison, for the other five F-GSLs, uptake was detected by at least one of the three GTRs, and they were either not detected in mock oocytes or were detected at very low levels (**GSL-A-Fluorescein** and **GSL-B-Fluorescein**) (Figure 2). These five F-GSL were deemed suitable for transport assays and were included in the experiment where uptake was detected by fluorescence measurements on oocyte homogenates (see below).

GTR1/2/3 are all characterized as active GSL transporters, which means that they can accumulate their GSL substrates against a concentration gradient [7,8]. Analyses of the LCMS-based transport assays show that GTR1 and GTR2 could transport **GSL-B-NBD**, **GSL-B-DNS**, and **GSL-A-Dansylamide** against the concentration gradient. GTR1 and GTR2 appear both to import **GSL-B-NBD** and **GSL-B-DNS** to similar levels, whereas GTR1 was several folds more efficient at importing **GSL-A-Dansylamide** compared to GTR2 (Figure 2). In comparison, GTR2 was more efficient at importing **GSL-B-Fluorescein** compared to GTR1; however, it did not accumulate **GSL-B-Fluorescein** against the concentration gradient, making **GSL-B-Fluorescein** inferior to the **GSL-B-NBD**, **GSL-B-DNS** and **GSL-A-Dansylamide** substrates. Of these four F-GSL, GTR3 was only able to import **GSL-B-NBD**. The GTR3-mediated import of **GSL-B-NBD** was active, albeit at three-fold lower levels compared to GTR1 and GTR2. Lastly, we saw a reciprocal uptake pattern for **GSL-A-Fluorescein**, which was imported by GTR3 and not by GTR1 or GTR2 (Figure 2).

After demonstrating GTR-mediated uptake of the five F-GSL, namely **GSL-A-Fluorescein**, **GSL-B-Fluorescein**, **GSL-B-NBD**, **GSL-B-DNS**, and **GSL-A-Dansylamide**, we investigated whether their uptake could be detected using fluorescence measurements of oocyte homogenates on a fluorescence plate reader. Similar to the LCMS-based assays, GTR-expressing oocytes were exposed to a 50 μM concentration of each of the five F-GSL at pH 5 for 1 h. As mentioned above, the same experiment was conducted three different times using different oocyte batches. With two exceptions, the results obtained by the fluorescence measurements matched the results obtained by LCMS-based detection (Figure 3). The two differences were both seen in assays using fluorescein-linked GSL. In contrast, to the first experiment, **GSL-B-Fluorescein** levels were now only significantly different from mock in GTR2-expressing oocytes (Figure 3), whereas significant uptake was detected in both GTR1- and GTR2-expressing oocytes in the LCMS-based experiment (Figure 2). For **GSL-A-Fluorescein**, significant uptake was now detected in both GTR3 and GTR2-expressing oocytes, whereas previously, we only saw an uptake in GTR3-expressing oocytes. For both of these aliphatic and benzylic **GSLs** bearing **fluorescein**, the absolute level of uptake was very low and was bordering detection limits for the **GSL-A-Fluorescein**. Problematically, it was seen that homogenates from oocytes that had not been exposed to any F-GSL gave a strong fluorescent signal when excited by the wavelength used to excite fluorescein. This signal was subtracted from the measurements on samples wherein GTR-expressing oocytes had been exposed to **GSLs** bearing **fluorescein**. Although uptake was detected for both aliphatic and benzylic GSL-Fluoresceins, the subtraction of a high background signal introduces an element of uncertainty that can explain the slight differences in uptake between the two experiments. Together, these data indicate that the fluorescein-linked GSLs are not as useful as substrates.

### 2.3. Natural Glucosinolate Glucoerucin Is in Competition with the Fluorescent Glucosinolates

Lastly, we sought to investigate whether the artificial F-GSLs are likely transported via the same mechanism as natural GSLs. Ideally, an estimation of Km of each GTR towards each of the three F-GSLs is needed. However, the limited amount of F-GSL available did not permit such investigations using electrophysiological measurements, which require high amounts of substrate. Additionally, although possible, Km estimation using LCMS or fluorescence-based measurements remains imprecise as both represent cumulative assays where one can only approximate initial transport rates. As an alternative, we challenged the uptake of each of the three F-GSLs with an equimolar concentration of Glucoerucin (4-methylthiobutyl glucosinolate, **4MTB**, Figure 1). This is a natural GSL towards which GTR1 and GTR2 exhibit an approximate Km of ~25 μM in oocytes [7], whereas GTR3 only transports **4MTB** passively and with low affinity [8].

Similar to the previous experiments, GTR-expressing oocytes were incubated at pH 5 for 1 h. The external buffer now contained either 50 μM **4MTB** alone, 50 μM of either of the three F-GSLs alone, or an intended equimolar concentration of **4MTB** and each of the respective F-GSLs individually (i.e., **4MTB** + **GSL-B-NBD**, **4MTB** + **GSL-B-DNS** or **4MTB + GSL-A-Dansylamide**).

The homogenate of the oocytes was divided into two parts. One part was analyzed on the plate reader, and the other half was analyzed by LCMS. The data interpretations presented in the following are corroborated by both detection methods.

As expected, when exposed to **4MTB** alone, GTR1 and GTR2 expressing oocytes imported **4MTB** actively and to similar levels. However, when **4MTB** was mixed with either of the three F-GSL, the level of **4MTB** uptake was reduced by approximately 50%. GTR1 and GTR2 also imported each individual F-GSL actively, and this uptake level was reduced by approx. 60% when mixed with an equimolar concentration of **4MTB** (Figure 4).

These results indicate that the natural and F-GSL are likely competing for the same substrate binding sites within the GTRs and that the fluorophores attached do not render the F-GSL as inferior substrates. For GTR3, we only analyzed the **GSL-B-NBD**, since GTR3 does not transport the other F-GSL (Figure 2 and Figure 3). As seen previously, GTR3 only imported **4MTB** up to–but not above the concentration in the external medium. This uptake of **4MTB** was significantly lowered when **GSL-B-NBD** was added in equimolar concentrations and vice versa (Figure 4).

## 3. Discussion

Since their discovery, GTRs have been functionally expressed and characterized in several heterologous host organisms via different assays. Most studies have used *Xenopus* oocytes as expression hosts and have characterized transport activity directly by measuring imported glucosinolate levels via LCMS or indirectly by measuring the currents elicited by TEVC electrophysiology. LCMS has also been used to measure GTR-mediated glucosinolate import in other heterologous hosts, including yeast [25], cotton cells [26], and insect cells (unpublished data) [27]. In one example, glucosinolate import was detected by monitoring the influx of radiolabeled glucosinolates that were generated by feeding radiolabeled tyrosine to CYP79A1 overexpressing *Arabidopsis* plants [7,28].

A clear advantage of the fluorescence-based GSL uptake assays presented here is that they are rapid and that detection can be accomplished by fluorescence microscopy, fluorescence plate reading, or flow cytometry. Additionally, single-cell fluorescence-based transport assays allow GTR-mutant library screening using fluorescence-activated cell sorting (FACS). For these purposes, the azide-glucosinolate precursors are produced in abundance and can be used to generate more of the three best-performing F-GSL described in this study **GSL-B-NBD**, **GSL-B-DNS**, and **GSL-A-Dansylamide**. In addition, they can be used to generate other F-GSLs with different fluorophores.

Oocyte homogenates did not emit fluorescence at the emission wavelengths of Dansylamide and NBD (Excitation at 335 nm and 465 nm, Emission at 518 nm and 560 nm, respectively), which resulted in very low background signal in mock oocytes. In comparison, the homogenates emitted a strong signal for the settings for fluorescein (Excitation at 492 nm and Emission at 524 nm), which renders the Fluorescein-linked GSLs less suitable for transport assays using *Xenopus* oocytes. Thus, it is advisable to investigate the spectral properties of heterologous expression hosts prior to selecting which F-GSL to use.

So far, GTR1 and GTR2 have been shown to transport glucosinolates with near similar preference irrespective of the structure of the glucosinolate side chain [7,25]. In comparison, GTR3 displays a strong preference for tryptophan-derived glucosinolates [8]. The ability of all GTRs to transport glucosinolates with fluorophores attached to the side chain shows that the GTR promiscuity towards the side chain extends beyond natural glucosinolates.

Rhodamine is by far the largest of the five tested fluorophores. Hence, the lack of transport by any of the GTRs likely indicates that we exceeded the structural constraints of the substrate binding pocket. However, we cannot exclude that the transporters may exhibit different kinetics towards the different F-GSLs. Therefore, it is possible that the **GSL-A-Rhodamine** may be transported by the GTRs if applied at higher concentrations. BODIPY is a lipid-binding fluorophore, which is typically used to quantify levels of neutral lipids in biological samples [29,30]. The high background levels seen in the transport assays likely reflect the binding of the fluorophore to the oocyte membrane, rendering **GSL-A-BODIPY** unsuitable for transport assays. To explore whether GSL linked to BODIPY can be used as substrates by GTRs, it would be necessary to use a modified non-lipid binding BODIPY as a fluorophore [31].

We did not measure the amount of GTR protein expressed in this study. Differences in GTR protein expression could therefore, in principle, explain the variation in uptake of different F-GSLs. However, we have shown in earlier studies that the oocytes generally express the different GTRs to very similar levels [8]. Accordingly, it is more likely that the different uptake levels reflect variations in substrate preference.

Kinetic values such as Km were not determined in this study. This is mainly due to a lack of sufficient amounts of F-GSL to carry out these investigations. In addition, we noticed that for the Fluorescein linked F-GSL, increasing external concentrations (beyond 100 μM) increased background levels in mock oocytes to the extent that masked uptake by GTR. However, the ability of GTR1 and GTR2 to accumulate the Dansylamide and NBD-linked F-GSL’ actively (above external media level) combined with the reduction of **4MTB** uptake in equimolar competition assays indicates that these F-GSL are transported by the same mechanism as natural GSL and with similar affinity. This is further supported by the lower uptake by GTR3 of these F-GSL, which mimics the inherent difference in substrate preference of GTR1/GTR2 versus that of GTR3 [8]. Accordingly, the F-GSL provides the prerequisites for conducting high-throughput screening of mutant libraries in studies aimed at unraveling the structural basis underlying substrate specificity in the GTRs. In this context, it is noteworthy that GTR1 and GTR2 have so far been shown to have almost completely overlapping substrate preference [7,8,25]. Here, the similar uptake of **GSL-B-DNS** and **GSL-B-NBD** but varying uptake of **GSL-A-Dansylamide** by GTR1 and GTR2, respectively, indicates that they harbor distinct substrate preferences. Whether this also extends to natural GSL remains to be investigated.

In addition, owing to their presence in *Arabidopsis thaliana*, the GTRs have emerged as a model system for studying the transport of specialized metabolites in planta. In this context, in vivo feeding of F-GSL could be utilized to identify apoplastic barriers at cellular resolution. Similar investigations could be conducted in specialized insects such as flea beetles that have evolved distinct glucosinolate transporters that enable the beetle to take up GSL from host plants and use it for their own defense [27]. However, it is not known whether the F-GSL are accepted as substrates by the flea beetle glucosinolate transporters that belong to a different transporter family.

## 4. Materials and Methods

### 4.1. Synthesis of the Aliphatic Glucosinolates: GSL-A-BODIPY, GSL-A-Dansylamide, and GSL-A-Rhodamine

#### 4.1.1. General Methods

Unless otherwise noted, all reagents were purchased from commercial suppliers and used without further purification. (*N,N*-Dimethylformamid (DMF): *Acros Organics*, puriss., extra dry, over molesieve (water ≤ 0.005%), Pyridin (Pyr): *Acros Organics*, puriss., extra dry, over molesieve (water ≤ 0.005%), Dimethylsulfoxid (DMSO): *Acros Organics*, puriss., extra dry, over mol sieve (water ≤ 0.005%), Methanol (MeOH): *Acros Organics*, puriss., extra dry (water ≤ 0.005%)). Moisture-sensitive reactions were performed under an argon atmosphere in dried glassware. Dry dichloromethane, diethyl ether, toluene, and tetrahydrofuran for moisture-sensitive reactions have been taken from an MB-SPS-800 (MBraun) solvent purifications system and stored under argon. All solvents used for workup and purification were of HPLC grade. Reactions were monitored by TLC, LCMS, or NMR. The solution of compounds in organic solvents was concentrated using rotary evaporators at a water bath temperature of max. 30 °C. Solvent residues were removed in a high vacuum at a pressure of appr. 10^−2^ mbar. Unless otherwise noted, solvents were degassed either by a continuous Argon flow over a minimum of 15 min or using the Freeze-Pump-Thaw technique [32]. Flash chromatography [33] was conducted using appropriate glass columns filled with silicagel (Merck Millipore, Geduran^®^ Si60, 1.11567.9025, 40–63 μm) or using the Biotage Select^®^ chromatography system with a DAD detector and cartridges packed with silicagel (Merck Millipore, Geduran^®^ Si60, 1.11567.9025, 40–63 μm) using a Cartridger^®^ C-670 from the company BüchiLabortechnik AG, Flawil, Switzerland. Preparative reversed-phase high-pressure liquid chromatography (prep. HPLC RP) was performed on either a Hypersil GOLD C18 RP-column (Part No. 25005-259270), 5 μm, 250 mm × 21.2 mm (10 mL/min) or a Hypersil GOLD C18 RP-column (Part No. 25005-259070A), 5 μm, 250 mm × 10.0 mm (5 mL/min) each equipped with a guard column of the same material using a Thermo Fisher Scientific (Waltham, MA, USA) Dionex Ultimate 3000 HPLC system. Eluents, gradients, and additives are given in parentheses. As eluents, HPLC grade acetonitrile and water (VWR Chemicals, Darmstadt, Germany, HPLC grade) with or without 0.1% of TFA (Carl Roth,, Karlsruhe, Germany, 6957.1, 99.9%) or buffer added were used. Appropriate reaction mixtures were filtered through CHROMAFIL^®^ PET-45/15 MS filters (45 μm) before being injected. Product-containing fractions were combined and diluted with dest. H_2_O (min. 1:1/solvent:H_2_O), frozen and lyophilized using a VaCo_2_^®^ Freeze dryer from Zirbus(Bad Grund, Germany, −80 °C, 0.05 mbar). Thin-layer chromatography (TLC) was performed on pre-coated glass plates (Merck, Darmstadt, Germany, TLC Silicagel 60 F_254_, 1.15341.0001, 2.5 × 7.5 cm), and components were visualized by observation under UV light (λ = 254 nm [UV^254^] or λ = 366 nm [UV^366^]) or visible light, treatment of developed plates in an iodine chamber or by treating the plates with TLC staining solutions (for preparation see list below) followed by heating. Eluent or eluent mixtures used are reported in parentheses.

CAM staining solution [CAM]: 1 g Ce(IV)(SO_4_)_2_, 2.5 g (NH_4_)_6_Mo_4_O_7_ in 100 mL 10% H_2_SO_4_Ninhydrin staining solution [Ninhydrin]: 1.5 g Ninhydrin in 100 mL *abs.* EtOH and 3.0 mL HOAc.

NMR spectra were recorded on Bruker AV-300, AVIII400, and AVIIIHD500 with a cryoprobe system at 293.15 K. ^1^H NMR spectra were recorded at 300 MHz, 400 MHz, and 500 MHz. ^13^C NMR spectra were recorded at 76 MHz, 100 MHz, and 126 MHz. Chemical shifts are reported in ppm relative to the solvent signal. Multiplicity is indicated as follows: s (singlet); bs (broad singlet); d (doublet); t (triplet); q (quartet); m (multiplet); dd (doublet of doublets), etc. For the processing of the raw data, the software MestReNova (Version 9.0.1-13254) from MestreLab Research S.L. was utilized. IR spectra were recorded on a Bruker Tensor 27 IR spectrometer with the ATR technique. Only the wave numbers of observed absorption peaks are given. Low-resolution mass spectrometry (LRMS) data were recorded using an LC-MS system consisting of an Accela HPLC (Thermo Scientific) equipped with an Accela photodiode array (PDA) Detector, Accela autosampler, and Accela 1250 pump which was coupled to an LTQ XL mass spectrometer (Thermo Scientific) for HPLC/HESI-MS analyses. Heated electrospray ionization was used with an enhanced scan range of 120 to 2000 amu. Gradient HPLC solvent programs consisted of LCMS-grade H_2_O, CH_3_CN, and 2% formic acid in H_2_O. An Agilent Zorbax Eclipse Plus C18 (3.5 μm, 2.1 × 150 mm) column was used, which was kept at 30 °C. The PDA detector was set to a scanning range from 190 to 600 nm with 1 nm wavelength steps. High-resolution mass spectrometry (HRMS) data were recorded on a Finnigan MAT 95 (EI, 70 eV) mass spectrometer and a Finnigan MAT 95 XL (ESI) mass spectrometer. UV-Vis spectroscopy data were recorded on a Cary 100 Bio (Varian). Fluorescence Emission Spectroscopy data were recorded on a Cary Eclipse (Varian).

#### 4.1.2. Synthesis of the GSL-A-N_3_ and GSL-A-BODIPY

The synthesis of **GSL-A-BODIPY** and **GSL-N_3_** was performed as outlined in Figure 3.

#### 4.1.3. Synthesis of GSL-A-Dansylamide

The synthesis of **GSL-A-Dansylamide** was performed as outlined in Figure 4.

5-(Dimethylamino)-*N*-(prop-2-yn-1-yl)naphthalene-1-sulfonamide.



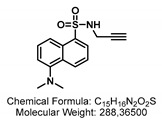



DiPEA (25.7 μL, 148 μmol, 2.0 eq) and propargylamine (5.2 μL, 81.6 μmol, 1.1 eq) were added to a solution of dansyl chloride (20 mg, 74 μmol, 1.0 eq) in dry CH_2_Cl_2_ (0.74 mL, 0.1 M) under Ar atm. at 0 °C. The reaction mixture was stirred for 18 h at 23 °C. The solvents were removed under reduced pressure, and the residue was purified by flash column chromatography through silica gel (EtOAc:Hexane/1:9 → 2:8 → 3:7) to yield Dansylpropargylamid (15.8 mg, 54.8 μmol, 74%) as a green oil.

TLC (EtOAc:Hex/3:7) R_f_: 0.4 [UV^254,366^]. ^1^H NMR (300 MHz, CDCl_3_) δ [ppm]: 8.57 (d, J = 8.5 Hz, 1H), 8.31–8.20 (m, 2H), 7.65–7.46 (m, 2H), 7.20 (d, J = 7.6 Hz, 1H), 4.87 (t, J = 6.1 Hz, 1H), 3.78 (dt, J = 6.1, 1.7 Hz, 2H), 2.90 (s, 6H). ^13^C-NMR (75 MHz, CDCl_3_) δ [ppm]: 134.4, 130.9, 130.1, 129.9, 128.7, 123.4, 115.4, 72.9, 45.6, 33.2. UV-Vis (c = 0.128 mg/10 mL in CH_2_Cl_2_). λ_max_ = 434 nm, ε_00_ = 7,727,414.8 M^−1^ cm^−1^. Fluorescence (c = 0.128 mg/10 mL in CH_2_Cl_2_, λ_Ex_ = 415 nm) λ_Em_ = 505 nm. The analytical data were in good accordance with prior published ones [34].

GSL-A-Dansylamide.



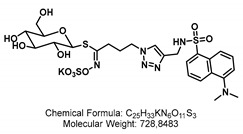



**GSL-A-N_3_** (5.8 mg, 13.18 μmol, 1.0 eq) and Dansylamide (3.8 mg, 13.18 μmol, 1.0 eq) were dissolved in a solvent mixture of DMSO (210 μL) and H_2_O (52 μL). TBTA (1.38 mg, 2.6 μmol, 0.2 eq), NaAsc (1.0 mg, 5.2 μmol, 0.4 eq), and Cu_2_SO_4_ (0.66 mg, 2.6 μmol, 0.2 eq) were added, and the mixture was stirred for 3 h at 23 °C, then diluted with 500 μL MeCN:H_2_O solution (30:70 + 1% TFA), and the precipitate was filtered off using a 45° μm Whatman^®^ filter. The filtrate was purified by preparative HPLC (HypersilGold, RP-18, MeCN:H_2_O/10:90 → 95:5 + 0.01% TFA in 45 min), and the product-containing fraction was lyophilized to obtain **GSL-A-Dansylamide** (1.7 mg, 2.33 μmol, 18%) as a fluffy white powder.

TLC (MeOH:CH_2_Cl_2_/2:8) R_f_: 0.44 [UV^254,366^, VIS (neon yellow)]. MS (EI) [*m*/*z*]: 689.3, 689.1 calculated for [C_25_H_33_N_6_O_11_S_3_]^−^. ^1^H NMR (500 MHz, Methanol-*d4*) δ [ppm]: 8.48 (d, J = 8.5 Hz, 1H), 8.45 (dt, J = 8.5, 1.0 Hz, 1H), 8.16 (dd, J = 7.4, 1.1 Hz, 1H), 7.67 (d, J = 7.1 Hz, 0H), 7.65–7.57 (m, 2H), 7.20 (s, 1H), 4.65 (d, J = 9.8 Hz, 1H), 4.18 (s, 2H), 4.06 (t, J = 7.1 Hz, 2H), 3.72 (dd, J = 12.2, 1.5 Hz, 1H), 3.53–3.48 (m, 1H), 3.31–3.26 (m, 1H), 3.25 (s, 1H), 3.20–3.15 (m, 2H), 3.14 (s, 7H), 2.55–2.36 (m, 2H), 1.88–1.80 (m, 2H). ^13^C-NMR (126 MHz, Methanol-*d4*) δ [ppm]: 160.7, 138.3, 131.2, 130.5, 129.0, 128.5, 126.6, 124.7, 118.9, 83.4, 82.2, 79.5, 74.1, 71.2, 62.8, 57.6, 57.5, 57.3, 50.0, 49.6, 47.4, 40.4, 38.7, 30.3, 28.3.

#### 4.1.4. Synthesis of GSL-A-Rhodamine

The synthesis of **GSL-A-Rhodamine** was performed as outlined in Figure 5.

2-(Prop-2-yn-1-yloxy)acetic acid.



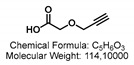



Propargyl alcohol (115 μL, 2.0 mmol, 1.0 eq) was added to a solution of NaH (80 mg, 2.0. mmol, 1.0 eq) in dry THF:DMF (5:1, 8 mL, 0.25 M) at 0 °C. The reaction mixture was stirred at 23 °C for 40 min before methyl 2-bromoacetate (306 mg, 2.0 mmol, 1.0 eq) was added at 0 °C and stirred at 23 °C for 5 h. The reaction mixture was diluted with Et_2_O (20 mL) and washed with H_2_O (20 mL) and brine (20 mL), dried over Na_2_SO_4_, filtered, and concentrated under reduced pressure.

The residue was dissolved in THF (2 mL, 1.0 M), and a 1.0 M solution of sodium hydroxide (2.6 mL, 2.6 mmol, 1.3 eq) was added at 23 °C and stirred for 3 h. Quenched with H_2_O (10 mL) and washed with Et_2_O (2 × 20 mL). The aqueous layer was acidified and extracted with Et_2_O (3 × 20 mL), dried over Na_2_SO_4_, filtered, and concentrated under reduced pressure. The residue was purified by flash column chromatography through silica gel (EtOAc:Hex/1:9 → 2:8 → 3:7), yielding 2-(prop-2-yn-1-yloxy)acetic acid as a colorless oil (0.142 g, 1.24 mmol, 62%).

^1^H NMR (300 MHz, CDCl_3_) δ [ppm]: 10.67 (s, 1H), 4.24 (d, J = 2.4 Hz, 2H), 4.18 (s, 2H), 2.47 (t, J = 2.4 Hz, 1H), 1.98 (s, 1H). ^13^C-NMR (75 MHz, CDCl_3_) δ [ppm]: δ 174.5, 78.2, 75.9, 65.6, 58.2. The analytical data were in good accordance with prior published ones [35].

*N*-(6-(diethylamino)-9-(2-(piperazine-1-carbonyl)phenyl)-9,9a-dihydro-3H-xanthen-3-ylidene)-*N*-ethylethanaminium 2,2,2-trifluoroacetate salt



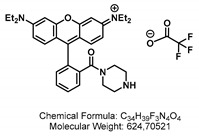



A 2.5 M solution of sodium hydroxide (0.6 mL, 1.5 mmol, 1.0 eq) was added to a solution of piperazine (0.32 g, 3.75 mmol, 2.5 eq) in *tert*-butanol (3 mL) at 23 °C. A solution of Boc-anhydride (0.33 g, 1.5 mmol, 1.0 eq) in *tert*-butanol (2 mL) was slowly added over a period of 20 min, and the reaction mixture was vigorously stirred for 1 h at 23 °C. The volatile solvent was removed under reduced pressure before the aqueous layer was extracted with CH_2_Cl_2_ (3 × 15 mL). The combined organic phases were dried over Na_2_SO_4_, filtered, and concentrated under reduced pressure.

Rhodamine B (0.72 g, 1.5 mmol, 1.0 eq), EDC HCl (0.37 g, 1.95 mmol, 1.3 eq), and DiPEA (0.52 mL, 3 mmol, 2.0 eq) were added to a solution of the residue in dry CH_2_Cl_2_ (4 mL, 0.4 M) under Ar atm, and the reaction mixture was stirred for 20 h at 23 °C. The reaction mixture was concentrated under reduced pressure. The residue was purified by flash column chromatography through silica gel (MeOH:CH_2_Cl_2_/2:98 → 5:95 → 1:9) yielding *N*-(9-(2-(4-(*tert*-butoxycarbonyl)piperazine-1-carbonyl)phenyl)-6-(diethylamino)-3*H*-xanthen-3-ylidene)-*N*-ethylethanaminium as an iridescent deep purple powder.

TFA (1.6 mL, 1.0 M) was added to a solution of the product in CH_2_Cl_2_ (1.6 mL, 1.0 M) at 0 °C and stirred for 90 min before the mixture was concentrated under reduced pressure yielding *N*-(6-(diethylamino)-9-(2-(piperazine-1-carbonyl)phenyl)-9,9a-dihydro-3H-xanthen-3-ylidene)-*N*-ethylethanaminium 2,2,2-trifluoroacetate salt as an iridescent deep purple powder (0.94 g, 1.5 mmol, quant.).

TLC (MeOH:CH_2_Cl_2_/1:9) R_f_: 0.22 [UV^254,366^, VIS (neon pink)]. IR (ATR) [cm^−1^]: 2983, 2341, 1740, 1686, 1645, 1591, 1475, 1415, 1343, 1278, 1184, 1128, 1076, 1010, 826, 687, 598, 568. ^1^H NMR (300 MHz, CDCl_3_) δ [ppm]: 7.68 (dd, J = 5.7, 3.3 Hz, 2H), 7.54–7.49 (m, 1H), 7.35 (dd, J = 5.7, 3.3 Hz, 1H), 7.22 (d, J = 9.5 Hz, 2H), 6.96 (s, 2H), 6.80 (s, 2H), 3.65 (q, J = 7.2 Hz, 8H), 3.41–3.14 (m, 9H), 1.42 (s, 3H), 1.40 (s, 10H), 1.32 (t, J = 7.1 Hz, 13H). ^13^C-NMR (75 MHz, CDCl_3_) δ [ppm]: 167.7, 157.8, 155.8, 154.5, 135.2, 132.1, 130.6, 130.3, 127.7, 113.8, 96.6, 80.7, 53.6, 46.3, 31.1, 28.6, 28.4, 12.8.

*N*-(6-(diethylamino)-9-(2-(4-(2-(prop-2-yn-1-yloxy)acetyl)piperazine-1-carbonyl)phenyl)-3*H*-xanthen-3-ylidene)-*N*-ethylethanaminium 2,2,2-trifluoroacetate.



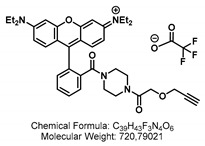



CDI (0.32 g, 2.0 mmol, 1.2 eq) was added to a solution of 2-(prop-2-yn-1-yloxy)acetic acid (0.228 g, 2.0 mmol, 1.2 eq) in dry DMF (13 mL, 0.1 M) under Ar atm, and the reaction mixture was stirred for 1 h at 23 °C. The mixture was added to a solution of *N*-(6-(diethylamino)-9-(2-(4-(2-(prop-2-yn-1-yloxy)acetyl)piperazine-1-carbonyl)phenyl)-3*H*-xanthen-3-ylidene)-*N*-ethylethanaminium 2,2,2-trifluoroacetate (0.81 g, 1.3 mmol, 1.0 eq) in DMF (2 mL) and stirred for 18 h at 23 °C. The solvent was removed under reduced pressure. The residue was purified by flash column chromatography through silica gel (MeOH:CH_2_Cl_2_/0:1 → 5:95 → 1:9) yielding *N*-(6-(diethylamino)-9-(2-(4-(2-(prop-2-yn-1-yloxy)acetyl)piperazine-1-carbonyl)phenyl)-3*H*-xanthen-3-ylidene)-*N*-ethylethanaminium 2,2,2-trifluoroacetate as an iridescent gold powder (0.60 g, 0.99 mmol, 76%).

TLC (MeOH:CH_2_Cl_2_/1:9) R_f_: 0.22 [UV^254, 366^, VIS (neon pink)]. IR (ATR) [cm^−1^]: 2979, 2344, 1692, 1640, 1590, 1476, 1416, 1343, 1275, 1182, 1129, 1075, 1008, 922, 824, 684, 598, 554. HRMS (EI) [*m*/*z*]: 607.32789, 607.32788 calculated for [C_37_H_43_N_4_O_4_]^+^ err [ppm] 0.016. ^1^H NMR (300 MHz, CDCl_3_) δ [ppm]: 7.90–7.81 (m, 1H), 7.71–7.63 (m, 2H), 7.54 (d, *J* = 3.6 Hz, 1H), 7.32 (dd, *J* = 5.9, 3.2 Hz, 1H), 7.24–6.67 (m, 5H), 4.21 (d, *J* = 2.7 Hz, 4H), 3.73–3.54 (m, 10H), 3.45 (d, *J* = 29.6 Hz, 10H), 1.30 (t, *J* = 7.1 Hz, 17H). ^13^C-NMR (75 MHz, CDCl_3_) δ [ppm]: 181.4, 167.8, 161.2, 157.8, 155.8, 130.4, 127.7, 96.5, 68.3, 58.6, 46.2, 18.3, 12.7, 7.9, 7.5, 7.3, 7.2, 7.1, 7.0, 6.9, 6.8, 6.7. ^19^F-NMR (282 MHz, CDCl_3_) δ [ppm]: −75.37. UV-Vis (c = 0.056 mg/10 mL in CH_2_Cl_2_). Λ_max_ = 563 nm, ε_00_ = 72,465,137.5 M^−1^ cm^−1^. Fluorescence (c = 0.056 mg/10 mL in CH_2_Cl_2_, λ_Ex_ = 545 nm) λ_Em_ = 585 nm.

GSL-A-Rhodamine.



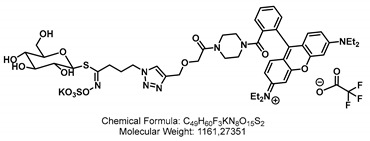



**GSL-A-N_3_** (17.2 mg, 39 μmol, 1.0 eq) and Rhodamine **4** (28 mg, 39 μmol, 1.0 eq) were dissolved in a solvent mixture of DMSO (624 μL) and H_2_O (156 μL). TBTA (4.1 mg, 7.8 μmol, 0.2 eq), NaAsc (3.1 mg, 15.6 μmol, 0.4 eq), and Cu_2_SO_4_ (1.9 mg, 7.8 μmol, 0.2 eq) were added, and the mixture was stirred for 2 h at 23 °C, then diluted with 100 μL MeCN:H_2_O solution (30:70 + 1% TFA) and the precipitate was filtered off using a 45°μm Whatman^®^ filter. The filtrate was purified by preparative HPLC (HypersilGold, RP-18, MeCN:H_2_O/10:90 → 95:5 + 0.01% TFA in 45 min), and the product-containing fraction was lyophilized to obtain **GSL-A-Rhodamine** (16.4 mg, 14.1 μmol, 36%) as a purple amorph solid.

TLC (MeOH:CH_2_Cl_2_/2:8) R_f_: 0.4 [UV^254,366^, VIS (neon pink)]. HRMS (EI) [*m*/*z*]: 1009.38057, 1009.37940 calculated for [C_47_H_61_N_8_O_13_S_2_]^+^ err [ppm] 1.16. ^1^H NMR (300 MHz, Methanol-*d4*) δ [ppm]: 8.25 (d, J = 45.3 Hz, 1H), 7.83–7.68 (m, 4H), 7.51 (dd, J = 6.1, 2.5 Hz, 1H), 7.30 (dt, J = 9.5, 2.0 Hz, 2H), 7.11 (dd, J = 9.6, 2.5 Hz, 1H), 7.01–6.93 (m, 2H), 4.80 (d, J = 9.7 Hz, 1H), 4.72–4.48 (m, 3H), 4.23 (s, 1H), 3.84 (d, J = 12.2 Hz, 1H), 3.71 (q, J = 7.2 Hz, 8H), 3.57 (d, J = 16.8 Hz, 1H), 3.49–3.36 (m, 1H), 3.21 (d, J = 8.6 Hz, 1H), 2.66 (d, J = 11.7 Hz, 2H), 2.37 (s, 2H), 1.32 (t, J = 7.0 Hz, 13H). ^13^C-NMR (76 MHz, Methanol-*d4*) δ [ppm]: 196.3, 169.6, 168.5, 159.2, 157.2, 133.2, 131.7, 131.4, 114.9, 97.3, 83.5, 82.2, 79.5, 74.2, 64.8, 46.9, 12.9. ^19^F-NMR (282 MHz, Methanol-*d4*) δ [ppm]: −75.81.

### 4.2. Synthesis of the Aliphatic Glucosinolate: GLS-A-Fluorescein

#### 4.2.1. General Methods for GLS-A-Fluorescein, GSL-B-DNS, GSL-B-NBD and GSL-B-Fluorescein

Commercial solvents and reagents were purchased from SigmaAldrich, Acros, Alfa-Aesar, TCI, Carbosynth, and Fluorochem suppliers and were used without further purification unless otherwise stated. Anhydrous solvents were dried by standard methods: DCM was distilled over P_2_O_5_, THF and acetonitrile were purified with a dry station GT S100 immediately prior to use; dried methanol from ACROS ORGANICS, *N*,*N*-dimethylformamide and dioxane were dried over molecular sieves; pyridine and triethylamine were dried over potassium hydroxide. Molecular sieves were activated by heating overnight in an Erscem oven at 500 °C. For the anhydrous reactions, all glassware was dried in an oven overnight (100 °C), then removed and cooled down to rt under argon flow. Argon flow was dried with a solution of conc. H_2_SO_4_ solution and CaCl_2_. Flash silica column chromatography was performed on silica gel 60 N (spherical neutral, 40–63 μm, Merck, Darmstadt, Germany) and column chromatography on the C-18 reverse phase was performed using a Reveleris^®^ flash chromatography system. The reactions were monitored by thin layer chromatography (TLC) on silica gel 60F254 precoated aluminum plates. Compounds were visualized under UV light (λ = 254 nm or λ = 365 nm) and by charring with a 10% H_2_SO_4_ ethanolic solution. Infrared spectra (ATR) were recorded on a PerkinElmer PARAGON 1000 PC instrument, and values were reported in cm^−1^. Mass Spectra were performed on a PerkinElmer Sciex API 300 mass spectrometer (low resolution) and a Bruker MaXis Q-Tof (High resolution) from the “Federation de Recherche” ICOA/CBM FR2708 platform in the electrospray ionization (ESI) mode. Optical rotation was measured at 20 °C into sodium light using a Jasco P-2000 polarimeter with a quartz tank with a path length of 1 cm; values are given in deg.dm^−1^.g^−1^ mL^−1^ with concentrations reported in g/100 mL. ^1^H NMR and ^13^C NMR were recorded with Bruker Avance II 250 MHz or an Avance III HD Nanobay 400 MHz spectrometer. CD_3_OD, D_2_O, and mainly CDCl_3_, with tetramethylsilane as an internal reference, were used as deuterated solvents. Acetone was added to D_2_O NMR samples as an internal reference for carbon NMR. Assignments were based on DEPT 135 sequence, homo- and heteronuclear 2D correlations. Chemical shifts were reported in parts per million (ppm). Coupling constants (J) are reported, expressed in Hertz (Hz), and rounded to the nearest 0.5 Hz; splitting patterns are designated as b (broad), s (singlet), d (doublet), dd (doublet of doublet), ddd (doublet of doublet of doublet), t (triplet), dt (doublet of triplet), q (quartet), or m (multiplet). To simplify, the NMR attribution protecting group was abbreviated: acetyl group (OAc). The following solvents were abbreviated: DCM (dichloromethane), DMF (*N*,*N*-dimethylformamide), EA (ethyl acetate), MeOH (methanol), and PE (petroleum ether).

#### 4.2.2. Synthesis of GSL-A-Fluorescein

Synthesis of 6-azidohexanal oxime (Figure 6, **2**).

Sodium azide (18.5 g, 0.285 mol, 2.7 eq) was added to a suspension of 6-chloro-hexan-1-ol **1** (14.4 g, 0.105 mol, 1 eq) in water (116 mL), and the reaction mixture was heated at reflux for 19 h. It was then cooled down to rt and extracted 3 times with ethyl acetate. The combined organic phases were dried over MgSO_4_, filtered and evaporated to give the crude azide product as a colorless oil.

Trichloroisocyanuric acid (TCCA) (24.7 g, 0.106 mol, 1.05 eq) was added to a solution of crude 6-azido-1-hexanol in anhydrous DCM (160 mL) at 0 °C. A solution of 2,2,6,6-Tetramethylpiperidine 1-oxyl (TEMPO) (158 mg, 1.01 mmol, 0.01 eq) in 3 mL of DCM was then added dropwise. The reaction mixture was stirred at 4 °C during 10 min, then at rt for 30 min, then filtered through a pad of Celite. The filtrate was washed once with an aqueous saturated solution of sodium carbonate, once with water, once with a 1 M HCl solution, and then once with brine. The organic phase was finally dried over MgSO_4_, filtered, and the solvent was evaporated under reduced pressure. The crude product was then purified by silica gel column chromatography (PE/EA: 100/0 to 0/100) to yield the pure aldehyde (4.4 g, 29%) as a colorless oil.

6-azidohexanal (4.4 g, 31.2 mmol, 1 eq) was dissolved in a mixture of H_2_O/MeOH 3/7 (42 mL), then hydroxylamine hydrochloride (2.7 g, 38.9 mmol, 1.25 eq) and potassium carbonate (2.15 g, 3.8 mmol, 0.50 eq) were added. The reaction mixture was stirred at room temperature for 4 h; then, the solvent was evaporated under reduced pressure. The crude residue was then taken up with ethyl acetate and washed twice with water and then once with brine; after drying over MgSO_4_, the solvent was evaporated under reduced pressure to give the desired 6-azidohexanal oxime **2** as a 55/45 mixture of *E*/*Z* isomers, (4.4 g, 90%) as a yellow solid.

TLC (PE:EA/50:50) R_f_: 0.3. HRMS (ESI^+^): *m*/*z* calculated for C_6_H_12_N_4_O ([M + H]^+^): 157.1084, found 157.1081. *NMR Data fort the E/Z mixture: M = major, m = minor isomer:*
^1^H NMR (400 MHz, CDCl_3_) δ (ppm) = 8.51 (bs, 1H, OH_M_), 7.41 (t, ^3^*J* = 6.0 Hz, 1H, H-1_M_), 6.71 (t, ^3^*J* = 5.5 Hz, 0.8H, H-1_m_), 3.33–3.22 (m, 3.5H, CH_2_N_3*,*_), 2.43–2.36 (m, 1.6H, CH_2_CN_m_), 2.24–2.18 (m, 2H, CH_2_CN_M_), 1.66–1.37 (m, 11H, CH_2 m_,_M_). ^13^C NMR (100 MHz, CDCl_3_) δ (ppm) = 152.4 (C-1_m_), 151.8 (C-1_M_), 51.38 (CH_2_N_3m_), 51.36 (CH_2_N_3M_), 29.4 (CH_2_CN_M_), 28.64 (*CH*_2_CH_2_N_3_), 28.62 (*CH*_2_CH_2_N_3_), 26.5 (*CH*_2_(CH_2_)_2_N_3_), 26.3 (*CH*_2_(CH_2_)_2_N_3_), 26.1 (*CH*_2_CH_2_CN), 25.7 (*CH*_2_CH_2_CN), 24.88 (CH_2_CN_m_).

*(Z)-S-*(2,3,4,6-Tetra-*O*-acetyl-β-D-glucopyranosyl)-(5-azidopentano)thiohydroximate (Figure 6, **3**)

A sodium hypochlorite solution (12.5% active chlorine) (20 mL, 36.6 mmol, 3 eq) was added to a vigorously stirred solution of crude 6-azidohexanal oxime **2** (4.4 g, 28.2 mmol, 1 eq) in anhydrous DCM (140 mL), the color of the solution changed from yellow to blue before returning to yellow. The solution was then stirred for 20 min at room temperature. The organic phase was separated from the aqueous one and slowly added to a solution of 2,3,4,6-tetra-*O*-acetyl-1-thio-β-D-glucopyranose (5.13 g, 14.1 mmol, 0.5 eq) in 140 mL of anhydrous DCM at −10 °C under argon atmosphere, then anhydrous triethylamine (11.8 mL, 84.5 mmol, 3 eq) was added dropwise, and the solution was allowed to warm up to room temperature. After stirring for 2 h at rt, the reaction was quenched by the addition of water, and the aqueous phase was extracted twice with DCM. The combined organic layers were then washed twice with a 0.5 M aqueous HCl solution, dried over MgSO_4_, filtered, and the solvent was evaporated under reduced pressure. The crude product was purified by silica gel column chromatography (EP/EA 100/0 to 0/100) to give the thiohydroximate **3** as a yellow solid (5.4 g, 74%).

TLC (PE:EA/50:50) R_f_: 0.35. ${\left[\alpha \right]}_{D}^{20}$ = + 21.6 (*c* = 0.5 in MeOH). m.p.: 140 °C (decomposition). HRMS (ESI^+^): *m*/*z* calculated for C_20_H_30_N_4_NaO_10_S ([M + Na]^+^): 541.157485, found 541.156951.^1^H NMR (400 MHz, CDCl_3_) δ (ppm) = 8.93 (s, 1H, NOH), 5.28–5.20 (m, 1H, H-2), 5.09–4.99 (m, 3H, H-1, H-3, H-4), 4.21 (dd, ^2^*J*_6a-6b_ = 12.4 Hz, ^3^*J*_6a-5_ = 5.4 Hz, 1H, H-6a), 4.13 (dd, ^2^*J*_6b-6a_ = 12.3 Hz, ^3^*J*_6b-5_ = 2.3 Hz, 1H, H-6b), 3.76 (ddd, ^3^*J*_5-4_ = 10.0 Hz, ^3^*J*_5-6a_ = 5.4 Hz, ^3^*J*_5-6b_ = 2.4 Hz, 1H, H-5), 3.30 (t, ^3^*J* = 6.7 Hz, 2H, CH_2_N_3_), 2.59–2.41 (m, 2H, CH_2_CN), 2.08, 2.05, 2.04, 2.00 (4× s, 12H, CH_3_ Ac), 1.70–1.56 (m, 4H, CH_2 pent_), 1.48–1.38 (m, 2H, CH_2 pent_). ^13^C NMR (100 MHz, CDCl_3_) δ (ppm) = 170.7, 170.3, 169.5, 169.3 (C=O), 151.9 (C=N), 80.0 (C-1), 76.0 (C-5), 73.8 (C-2), 70.2 (C-3), 68.2 (C-4), 62.3 (CH_2_-6), 51.3 (CH_2_N_3_), 32.5 (CH_2_CN), 28.6, 26.5, 26.3 (CH_2 pent_), 20.73, 20.68, 20.61 (CH_3_ Ac). IR (neat) $\tilde{\nu }$ (cm^−1^) = 2160 (N_3_), 1740 (C=O), 1533, 987.

Pentyl Ac_4_GSL-A-N_3_, *(Z)-S-*(2,3,4,6-Tetra-O-acetyl-β-D-glucopyranosyl)-(5-azidopentano)thiohydroximate-*N*,*O*-sulfate potassium salt

The thiohydroximate **3** (3.5 g, 6.75 mmol, 1 eq) was dissolved in anhydrous DCM (84 mL), sulfur trioxide-pyridine complex (5.4 g, 33.7 mmol, 5 eq) was added, and the suspension was heated at reflux for 24 h. The reaction was then cooled down to 0 °C and quenched by the addition of a 0.5 M aqueous KHCO_3_ solution (6.8 g, 67.5 mmol, 10 eq) and stirred for 45 min at room temperature. The solvent was then evaporated under reduced pressure, and the residue was purified using silica gel column chromatography (ethyl acetate/methanol: 9/1) to give the sulfated compound **4** as a resin (3.6 g g, 84%).

TLC (EA:MeOH/80:20) R_f_: 0.4. ${\left[\alpha \right]}_{D}^{20}$ = + 14 (*c* = 0.68 in MeOH). HRMS (ESI^−^): *m*/*z* calculated for C_20_H_29_N_4_O_13_S_2_ ([M-K]^−^): 597.1178, found 597.1175. ^1^H NMR (400 MHz, CD_3_OD) δ (ppm) = 5.39 (d, ^3^*J*_3-4_ = 9.3 Hz, 1H, H-3), 5.37 (d, ^3^*J*_1-2_ = 10.1 Hz, 1H, H-1), 5.05 (t, ^3^*J*_4-3_ = ^3^*J*_4-5_ = 9.7 Hz, 1H, H-4), 4.98 (t, ^3^*J*_2-3_ = ^3^*J*_2-1_ = 9.6 Hz, 1H, H-2), 4.29 (dd, ^2^*J*_6a-6b_ = 12.5 Hz, ^3^*J*_6a-5_ = 5.3 Hz, 1H, H-6a), 4.21 (dd, ^2^*J*_6b-6a_ = 12.4 Hz, ^3^*J*_6b-5_ = 2.3 Hz, 1H, H-6b), 4.08 (ddd, ^3^*J*_5-4_ = 10.1 Hz, ^3^*J*_5-6a_ = 5.3 Hz, ^3^*J*_5-6b_ = 2.3 Hz, 1H, H-5), 3.41–3.35 (m, 2H, CH_2_N_3_), 2.77–2.70 (m, 2H, CH_2_CN), 2.07, 2.03, 2.02, 1.98 (4 x s, 12H, CH_3_ Ac), 1.83 (quin., ^3^*J* = 7.5 Hz, 2H, CH_2 pent_), 1.76–1.67 (m, 2H, CH_2 pent_), 1.62–1.52 (m, 2H, CH_2 pent_). ^13^C NMR (100 MHz, CD_3_OD) δ (ppm) = 172.2, 171.5, 171.2, 170.9 (C=O), 159.0 (C=N), 80.9 (C-1), 76.7 (C-5), 75.0 (C-3), 71.4 (C-2), 69.5 (C-4), 63.4 (CH_2_-6), 52.3 (CH_2_N_3_), 33.5 (CH_2_C=N), 29.6, 27.7, 27.2 (CH_2 pent_), 20.7, 20.54, 20.51 (CH_3_ Ac). IR (neat) $\tilde{\nu }$ (cm^−1^) = 2150 (N_3_), 1754 (C=O), 1533, 1266 (C-O), 1193 (S=O), 1003 (C-C).

**Pentyl GSL-A-N_3_**, *(Z)-S-*(β-D-glucopyranosyl)-(5-azidopentano)thiohydroximate-*N*,*O*-sulfate potassium salt

Potassium methoxide (132 mg, 1.88 mmol, 0.4 eq) was added to a solution of the acetylated compound **4** (3 g, 4.7 mmol, 1 eq) in anhydrous methanol (60 mL). The reaction mixture was stirred at room temperature for 6h then the solvent was evaporated under reduced pressure. The crude product was purified using Reveleris^®^ column chromatography on C-18 reverse phase (H_2_O/MeOH: 100/0 to 0/100) to give the glucosinolate **5** as a white resin (1.9 g, 86%).

TLC (EA:MeOH/80:20) R_f_: 0.1. ${\left[\alpha \right]}_{D}^{20}$ = + 32 (*c* = 0.6 in MeOH). HRMS (ESI^−^) *m*/*z* calculated for C_12_H_21_N_4_O_9_S_2_ ([M-K]^−^): 429.0755, found 429.0754. ^1^H NMR (400 MHz, CD_3_OD) δ (ppm) = 4.94 (d, ^3^*J*_1-2_ = 9.8 Hz, 1H, H-1), 3.74 (dd, ^2^*J*_6a-6b_ = 12.3 Hz, ^3^*J*_6a-5_ = 1.8 Hz, 1H, H-6a), 3.66 (dd, ^2^*J*_6b-6a_ = 12.2 Hz, ^3^*J*_6b-5_ = 5.3 Hz, 1H, H-6b), 3.45–3.23 (m, 6H, CH_2_N_3_, H-2, H-3, H-4, H-5), 2.80–2.63 (m, 2H, CH_2_CN), 1.79 (quint, ^3^*J* = 7.6 Hz, 2H, CH_2 pent_), 1.70–1.61 (m, 2H, CH_2 pent_), 1.55–1.46 (m, 2H, CH_2 pent_). ^13^C NMR (100 MHz, CD_3_OD) δ (ppm) = 162.2 (C=N), 83.6 (C-1), 82.2 (C-5), 79.4 (C-4 or C-3), 74.1 (C-2), 71.1 (C-3 or C-4), 62.6 (CH_2_-6), 52.3 (CH_2_N_3_), 33.5 (CH_2_CN), 29.5, 28.0, 27.3 (CH_2 pent_). IR (neat) $\tilde{\nu }$ (cm^−1^) = 3605 (OH), 2155 (N_3_), 1780, 1260 (C-O), 1545, 954, 789.

**GSL-A-Fluorescein**, *(Z)*-S-(β-D-glucopyranosyl)-[*N*-fluoresceinyl-*N*’pentyl)carbamothioyl) thiohydroximate-*N*,*O*-sulfate potassium salt

Triphenylphosphine (112 mg, 0.43 mmol, 2 eq) was added to a solution of pentylazido glucosinolate **5** (100 mg, 0.21 mmol, 1 eq) in a THF/Water (9/1) mixture (2.1 mL). The solution was heated at 50 °C for 5h, then fluorescein isothiocyanate (83 mg, 0.21 mmol, 1 eq) in DMSO (200 μL) was added, and the mixture was stirred at room temperature for a further 20h. The solvent was then evaporated under reduced pressure, and the crude product was purified using Reveleris^®^ column chromatography on C-18 reverse phase (H_2_O/MeOH: from 100/0 to 0/100) to give the desired product **6** as a red resin (100 mg, 56%).

TLC (EA:MeOH/80:20) R_f_: 0.1. ${\left[\alpha \right]}_{D}^{20}$ = + 20.2 (*c* = 0.45 in MeOH). HRMS (ESI^−^): *m*/*z* calculated for C_33_H_34_N_3_O_14_S_3_ ([M-K]^−^): 792.1208, found 792.1210. ^1^H NMR (400 MHz, CD_3_OD) δ (ppm) = 8.15 (bs, 1H, H_Ar_), 7.80–7.75 (m, 1H, H_Ar_), 7.17 (d, ^3^*J* = 8.2 Hz, 1H, H_Ar_), 6.97 (d, ^3^*J* = 8.9 Hz, 2H, H_Ar_), 6.68 (d, ^4^*J* = 2.3 Hz, 2H, H_Ar_), 6.63 (dd, ^3^*J* = 8.9 Hz, ^4^*J* = 2.3 Hz, 2H, H_Ar_), 4.90–4.88 (m, 1H, H-1), 3.88 (d, ^2^*J*_6a-6b_ = 12.1 Hz, ^3^*J*_6a-5_ = 2.1 Hz, 1H, H_6a_), 3.72–3.66 (m, 2H, CH_2_CN), 3.66 (dd, ^2^*J*_6b-6a_ = 12.1 Hz, ^3^*J*_6b-5_ = 5.8 Hz, 1H, H_6b_), 3.46- 3.40 (m, 1H, H-4), 3.38 (dd, ^3^*J*_5-6b_ = 5.6 Hz, ^3^*J*_5-6a_ = 2.1 Hz, 2H, H-5), 3.31–3.24 ( m, 2H, H-3, H-2), 2.77 (q, ^3^*J* = 7.3 Hz, 2H, CH_2 pent_), 1.82 (quint, ^3^*J* = 7.4 Hz, 2H, CH_2 pent_), 1.73 (quint, ^3^*J* = 7.3 Hz, 2H, CH_2 pent_), 1.55 (quint, ^3^*J* = 7.4 Hz, 2H, CH_2 pent_). ^13^C NMR (100 MHz, CD_3_OD) δ (ppm) = 161.9 (C_q_ Ar), 157.2 (C=N), 149.3 (C_q_ Ar), 142.0 (C_q_ Ar), 131.8 (CH Ar), 128.2 (CH Ar), 122.4 (CH Ar), 118.2 (CH Ar), 114.0 (C_q_ Ar), 103.8 (CH Ar), 83.6 (C-1), 82.2 (C-5), 79.5 (C-4), 74.2 (C-2), 71.2 (C-3), 62.7 (CH_2_-6), 44.8 (CH_2_CN), 33.2 (CH_2_N), 28.5, 26.9, 26.4 (CH_2 pent_). IR (neat) $\tilde{\nu }$ (cm^−1^) = 3223 (OH), 1198 (S=O), 1565 (C=C), 1103 (C-O), 684 (C_sp2_-H _Ar_).

### 4.3. Synthesis of the Benzylic Glucosinolates: GSL-B-DNS, GSL-B-NBD, and GSL-B-Fluorescein

The syntheses of **GSL-B-DNS**, **GSL-B-NBD** and **GSL-B-Fluorescein** was performed as outlined in Figure 7.

#### 4.3.1. General Procedure for Synthesis of GSL-B-NH_2_

Triphenylphosphine (55 mg, 0.21 mmol, 1.1 eq) was added to a solution of **GSL-B-N_3_** (100 mg, 0.19 mmol, 1 eq) in MeOH (5 mL). The reaction mixture was stirred at room temperature for 24h. Upon completion, 10 mL of distilled water was added, and a white precipitate was observed. After washing with ethyl acetate three times, the aqueous phase was concentrated under reduced pressure to afford the crude GSL-NH_2_.

#### 4.3.2. Synthesis of GSL-B-DNS

A solution of dansyl chloride (53 mg, 0.2 mmol, 1.1 eq) in DMF (2.5 mL) and Et_3_N (0.13 mL, 0.88 mmol, 5 eq) were added simultaneously and dropwise to a solution of **GSL-B-NH_2_** (90 mg, 0.18 mmol, 1 eq) in DMF (2.5 mL) at 0 °C. The reaction mixture was protected from light and stirred at room temperature overnight. Upon completion, the solvent was evaporated under reduced pressure, and the crude residue was purified using Reveleris^®^ column chromatography on C-18 reverse phase (H_2_O/MeOH 100/0 to 0/100) to give the desired product **GSL-B-DNS** as a pale green powder (72 mg, 55% yield over 2 steps).

${\left[\alpha \right]}_{D}^{20}$ = −16.5 (*c* = 1.0 in MeOH). HRMS (ESI^−^): *m*/*z* calculated for C_28_H_34_N_3_O_12_S_3_^−^ ([M-K]^−^): 700.1310, found 700.1312. ^1^H NMR (400 MHz, D_2_O) δ (ppm) = 8.38 (d, ^3^*J* = 8.6 Hz, 1H, H_Ar DNS_), 8.24 (d, ^3^*J* = 7.3 Hz, 1H, H_Ar DNS_), 8.20 (d, ^3^*J* = 8.7 Hz, 1H, H_Ar DNS_), 7.58 (t, ^3^*J* = 8.0 Hz, 1H, H_Ar DNS_), 7.50 (t, ^3^*J* = 8.2 Hz, 1H, H_Ar_), 7.23 (d, ^3^*J* = 7.6 Hz, 1H, H_Ar_), 7.02 (d, ^3^*J* = 8.4 Hz, 2H, H_Ar_), 6.17 (d, ^3^*J* = 8.4 Hz, 2H, H_Ar_), 4.71–4.63 (m, 1H, H-1), 4.02–3.91 (m, 2H, CH_2_-CN), 3.72–3.61 (m, 2H, CH_2_-6), 3.55–3.31 (m, 7H, H-2, H-3, H-4, CH_2_O, CH_2_N), 3.24 (ddd, ^3^*J*_5-4_ = 9.7 Hz, ^3^*J*_5-6a_ = 4.9 Hz, ^3^*J*_5-6b_ = 2.7 Hz, 1H, H-5), 2.80 (s, 6H, N(CH_3_)_2_). ^13^C NMR (100 MHz, D_2_O, internal acetone) δ (ppm) = 162.2 (C_q_ Ar), 156.5 (C=N), 150.6 (C_q_ Ar), 134.5 (C_q_ Ar), 130.1 (CH Ar _DNS_), 129.5 (CH Ar _DNS_), 129.0 (CH Ar _GSL_), 128.9 (CH Ar _DNS_), 128.8 (C_q_ Ar), 128.7 (C_q_ Ar), 127.2 (C_q_ Ar), 123.9 (CH Ar _DNS_), 118.9 (CH Ar _DNS_), 116.0 (CH Ar _DNS_), 114.4 (CH Ar _GSL_), 81.3 (C-1), 79.9 (C-5), 77.0 (C-2), 71.9 (C-3 or C-4), 68.8 (C-4 or C-3), 65.3 (CH_2_O), 60.4 (CH_2_-6), 48.9, 45.1 (CH_3_), 42.0 (CH_2_NH), 37.6 (*CH_2_*CN). IR (neat) $\tilde{\nu }$ (cm^−1^) = 3397 (O-H), 2960, 1611 (C=N), 1573 1511 (C=C), 1242 (C-N), 1141 (S=O), 1056 (C-O).

#### 4.3.3. Synthesis of GSL-B-NBD

NBD-Cl (39 mg, 0.2 mmol, 1.1 eq) and Et_3_N (0.13 mL, 0.88 mmol, 5 eq) were added to a solution of **GSL-B-NH_2_** (90 mg, 0.18 mmol, 1 eq) in MeOH (5 mL). The reaction mixture was protected from light and stirred at room temperature overnight. Upon completion, the solvent was evaporated, and the crude residue was purified using Reveleris^®^ column chromatography on C-18 reverse phase (H_2_O/MeOH 100/0 to 0/100) to give the desired product **GSL-B-NBD** as a red powder (84 mg, 71% yield over 2 steps).

${\left[\alpha \right]}_{D}^{20}$ = −16.5 (*c* = 1.0 in MeOH). HRMS (ESI^−^): *m*/*z* calculated for C_22_H_24_N_5_O_13_S_2_^−^ ([M-K]^−^): 630.0818, found 630.0819. ^1^H NMR (400 MHz, D_2_O) δ (ppm) = 8.21 (d, ^3^*J* = 8.7 Hz, 1H, H Ar _NBD_), 7.09 (d, ^3^*J* = 7.9 Hz, 2H, H Ar _GSL_), 6.73 (d, ^3^*J* = 7.9 Hz, 2H, H Ar _GSL_), 6.23 (d, ^3^*J* = 9.0 Hz, 1H, H Ar _NBD_), 4.69 (d, ^3^*J* = 8.9 Hz, 1H, H-1), 4.19 (bs, 2H, CH_2_O), 4.01–3.77 (m, 4H, CH_2_CN, CH_2_N), 3.76–3.56 (m, 2H, CH_2_-6), 3.52–3.26 (m, 3H, H-2, H-3, H-4), 3.25–3.19 (m, 1H, H-5). ^13^C NMR (100 MHz, D_2_O, internal acetone) δ (ppm) = 162.4 (C_q_ Ar), 156.9 (C=N), 146.8 (C_q_ Ar), 143.9 (C_q_ Ar), 138.6 (CH Ar _NBD_), 129.2 (CH Ar_GSL_), 127.6 (C_q_ Ar), 120.6 (Cq Ar), 114.7 (CH Ar_GSL_), 100.1 (CH Ar_NBD_), 81.3 (C-1), 79.8 (C-5), 76.9 (C-3/C-4), 71.9 (C-2), 68.8 (C-4/C-3), 65.9 (CH_2_O), 60.3 (CH_2_-6), 43.2 (CH_2_NH), 37.3 (CH_2_CN). IR (neat) $\tilde{\nu }$ (cm^−1^) = 3376 (O-H), 3045 (C_sp2_-H), 2940 (C_sp3_-H), 1584, 1510 (C=C), 1226 (C-N), 1188 (S=O), 1054 (C-O).

#### 4.3.4. Synthesis of GSL-B-Fluorescein

Fluorescein ITC (88 mg, 0.22 mol, 1 eq) was added to a solution of **GSL-B-NH_2_** (114 mg, 0.22 mol, 1 eq) in DMF (30 mL). The reaction mixture was stirred at room temperature overnight. Upon completion, the solvent was evaporated, and the crude residue was purified using Reveleris^®^ column chromatography on C-18 reverse phase (H_2_O/MeOH 100/0 to 0/100) to afford the desired product **GSL-B-Fluorescein** as an orange powder (135 mg, 66% over two steps).

${\left[\alpha \right]}_{D}^{20}$ = −9.7 (*c* = 1.0 in MeOH). HRMS (ESI^−^): *m*/*z* calculated for C_37_H_34_N_3_O_15_S_3_^−^ ([M-K]^−^): 856.1158, found 856.1171. ^1^H NMR (400 MHz, CD_3_OD) δ (ppm) = 8.18 (d, ^4^*J* = 2.0 Hz, 1H, H Ar _FG_, 7.77 (dd, ^3^*J* = 8.3, ^4^*J* = 2.0 Hz, 1H, H Ar _FG_), 7.34 (d, ^3^*J* = 8.3 Hz, 2H, H Ar _GSL_), 7.15 (d, ^3^*J* = 8.2 Hz, 1H, H Ar _FG_), 6.97 (d, ^3^*J* = 8.7 Hz, 2H, H Ar _GSL_), 6.57 (d, ^3^*J* = 8.8 Hz, 2H, H Ar _FG_), 6.68 (d, ^4^*J* = 2.3 Hz, 2H, H Ar _FG_), 6.56 (dd, ^3^*J* = 8.8 Hz, ^4^*J* = 2.4 Hz, 2H, H Ar _FG_), 4.57–4.47 (m, 1H, H-1), 4.26–4.14 (m, 3H, CH_2_O, (CH_2_CN)_a_), 4.03 (t, ^3^*J* = 5.4 Hz, 2H, CH_2_N), 3.98 (d, ^2^*J* = 15.8 Hz, 1H, (CH_2_CN)_b_), 3.85 (d, ^2^*J*_6a-6b_ = 12.1 Hz, 1H, H_6a_), 3.71–3.53 (m, 1H, H_6b_), 3.27–3.20 (m, 2H, H_2,_ H_5_), 3.19–3.09 (m, 3H, H_3_, H_4_, NH _FG_), 1.28 (t, ^3^*J* = 7.3 Hz, 1H, NH _GSL_). ^13^C NMR (100 MHz, CD_3_OD) δ (ppm) = 162.2 (C_q_ Ar _F_), 161.4 (C_q_ Ar _F_), 159.5 (C_q_ Ar), 157.9 (C=N), 153.6 (C_q_ Ar), 140.9 (C_q_ Ar), 129.7 (CH Ar _FG_), 129.3 (CH Ar _F_), 129.1 (CH Ar _GSL_), 128.3 (C_q_ Ar), 124.7 (CH Ar _F_), 119.2 (CH Ar _F_), 114.6 (CH Ar _GSL_), 113.5 (CH Ar _F_), 111.1 (C_q_ Ar _F_), 102.2 (CH Ar _F_), 81.4 (C-1), 80.8 (C-5), 77.9 (C-2), 72.8 (C-3), 69.8 (C-4), 65.9 (CH_2_O), 61.4 (CH_2_-6), 43.7 (CH_2_N) 37.5 (CH_2_CN). IR (neat) $\tilde{\nu }$ (cm^−1^) = 3310 (O-H), 3010 (C_sp2_-H), 2956 (C_sp3_-H), 1589, 1510 (C=C), 1241 (C-N), 1113 (S=O), 1056 (C-O).

### 4.4. DNA Constructs and cRNA Generation

Oocyte plasmids containing GTR1, GTR2, and GTR3 were obtained from previous publications [7,8] (Table 1).

Linearized DNA templates for cRNA generation were obtained by PCR. Templates contained the coding sequence and the surrounding *Xenopus* 5′- and 3′-UTRs from the pNB1. **Forward primer:** (5′-AATTAACCCTCACTAAAGGGTTGTAATACGACTCACTATAGGG-3′); **Reverse primer:** (5′-TTTTTTTTTTTTTTTTTTTTTTTTTTTTTATACTCAAGCTAGCCTCGAG-3′). PCR products were purified using E.Z.N.A^®^ PCR Cycle Pure kit (OMEGA bio-tek, Norcross, GA, USA) using the manufacturer’s instructions. Purified PCR products were in vitro transcribed with the mMessage mMachine T7 transcription kit (InVitrogen, Thermo Fisher Scientific, Waltham, MA, USA) using the manufacturer’s instructions. Quantification of yield measured with NanoDrop, RNA was diluted to a concentration of 500–550 ng/μL. Before use, the RNA was checked on an agarose gel.

### 4.5. Xenopus laevis Oocyte Transport Assays

*Xenopus laevis* transport assay was performed as described in [13]. Briefly, *X. laevis* oocytes stage V-VI were purchased from Ecocyte Biosciences. Oocytes are injected with 50.6 nL cRNA or nuclease-free water using a Nanoinject II (Drummond Scientific Company, Broomall, PA, USA). Glass capillars for the Nanoinject II were prepared with a needle puller, manually cut with surgical scissors, and filled with mineral oil. Injected oocytes were incubated for 3 days at 16 °C in buffer pH 7.4 (90 mM NaCl, 1 mM KCl, 1 mM MgCl_2_, 1 mM CaCl_2_, 5 mM HEPES).

Assay: Injected oocytes were pre-incubated in buffer pH 5 (90 mM NaCl, 1 mM KCl, 1 mM MgCl_2_, 1 mM CaCl_2_, 5 mM MES) for 2 min before transferring to compound containing buffer for 1 h. After 1 h, the oocytes were washed three times in buffer pH 7.4, followed by one wash in Milli-Q water. Media samples were taken after the addition of oocytes to compound containing buffer. For LCMS: 1 μL media + 1 oocyte, for platereader: 2 μL media + 2 oocytes. Media samples were treated as oocyte samples.

For LCMS: Single oocytes per sample were analyzed; each oocyte was homogenized in 62.5 μL 50% methanol containing the internal standard Sinigrin (1.25 μM) and stored minimum 1 h at −20 °C. The homogenized oocyte samples were pelleted at max speed for 10 min at 4 °C. Then, 50 μL of the supernatant was mixed with 75 μL Milli-Q water to a final methanol concentration of 20% before filtration through a 0.22 μm filter plate. Sample dilution: 156.25 (Considering the oocyte ~1 μL), final internal standard conc 0.5 μM.

For plate reader: Two oocytes per sample were analyzed; oocytes were homogenized in 110 μL miliQ water and stored at −20 °C until analysis. The homogenized oocyte samples were pelleted at max speed for 10 min at 4 °C, and 100 μL of the supernatant was analyzed on microplate reader. Sample dilution: 55 (Considering the oocyte ~1 μL).

#### Measurement on Microplate Reader

Measurements were conducted with a microplate reader from BioTek–Synergy H1 in black plates with the clear bottom from costar^®^. With the clear bottom, it was possible to read from the bottom. The settings used are displayed in Table 2.

### 4.6. Quantification with LCMS

Samples were subjected to analysis by liquid chromatography coupled with tandem mass spectrometry. The method was modified by Crocoll et al. [36], and parameters were adjusted and optimized to match the LC-MS/MS system in use. Briefly, chromatography was performed on a 1290 Infinity II UHPLC system (Agilent Technologies). Separation was achieved on a Kinetex XB-C18 column (100 × 2.1 mm, 1.7 μm, 100 Å, Phenomenex, Torrance, CA, USA). Formic acid (0.05%, *v*/*v*) in water and acetonitrile (supplied with 0.05% formic acid, *v*/*v*) were employed as mobile phases A and B, respectively. The elution profile for glucosinolates was: 0–0.2 min, 5% B; 0.2–3.5 min, 5–65% B; 3.5–4.2 min 65–100% B, 4.2–4.9 min 100% B, 4.9–5.0 min, 100–5% B and 5.0–6.0 min 5% B. The mobile phase flow rate was 400 μL/min. The column temperature was maintained at 40 °C. The liquid chromatography was coupled to an Ultivo Triplequadrupole mass spectrometer (Agilent Technologies, Santa Clara, CA, USA) equipped with a Jetstream electrospray ion source (ESI) operated in negative ion mode. The instrument parameters were optimized by infusion experiments with pure standards. The ion spray voltage was set to 4500 V. Dry gas temperature was set to 325 °C, and the dry gas flow to 13 L/min. The sheath gas temperature was set to 400 °C, and the sheath gas flow to 12 L/min. Nebulizing gas was set to 55 psi. Nitrogen was used as dry gas, nebulizing gas, and collision gas. Multiple reaction monitoring (MRM) was used to monitor precursor ion → fragment ion transitions. MRM transitions were determined by direct infusion experiments of reference standards. Detailed values for mass transitions can be found in Table 3. Both Q1 and Q3 quadrupoles were maintained at unit resolution. Mass Hunter Quantitation Analysis for QQQ software (Version 10, Agilent Technologies) was used for data processing. Linearity in ionization efficiency was verified by analyzing dilution series that were also used for quantification.

### 4.7. Data Analysis

All data analyses were performed using Rstudio. Concentrations from LCMS samples were calculated based on internal standard (Sinigrin, Prop-2-enyl GSL), and the response factor was generated based on standard curves. Concentrations from the plate reader were calculated based on standard curves slope–no fluorescent internal standard was used, hence no response factor. Outliers were removed automatically based on generated function in R that removes values outside of ±1.5*Inter quantile range. Statistical analysis and post hoc tests are stated in figure legends. All were performed using ANOVA function aov() in R (ANOVA tables in Appendix A), all but Figure 4 followed by TUKEY post hoc test. For Figure 4, the Dunnett Post hoc test was used.

LCMS data: Data from LCMS were calculated as described in [36], with a modification for the dilution factor. Briefly, the response factor for oocytes was determined by standard curves for the compound investigated and the internal standard: response factor (f) = slope internal standard/slope compound investigated. Concentration in samples was calculated: (Area of peak for compound/Area of peak for internal standard) * f * final internal standard concentration * dilution factor. Dilution factor: 156.25, final internal standard concentration: 0.5 μM.

Plate reader data: Concentration in samples was calculated: (Area of peak for compound/Slope standard curve) * dilution factor. Dilution factor: 55.

## 5. Conclusions

Seven artificial F-GSLs were generated by a combination of GSL precursors with five different fluorophores via two synthesis strategies. We demonstrate that the three F-GSLs, namely **GSL-B-NBD**, **GSL-B-DNS**, and **GSL-A-Dansylamide**, are transported actively by GTR1 and GTR2 and provide indications that they compete for a common binding site. Besides demonstrating their usefulness in heterologous GTR-based transport assays, the F-GSLs may be useful for unraveling the relationship between structure and function in the GTRs and possibly as tracers to monitor glucosinolate transport in vivo.

## Data Availability

The data presented in this study are available in Appendix A.

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
