# Peer review of "Artificial Fluorescent Glucosinolates (F-GSLs) Are Transported by the Glucosinolate Transporters GTR1/2/3"

_ijms, 2023, doi:10.3390/ijms24020920_

Round 1

Reviewer 1 Report

This manuscript focuses on the ability of plant glucosinolate transporters (GTRs) to transport synthetic, fluorescent glucosinolate analogs. This work is important for several reasons, it provides alternate substrates for the study of GTRs in plants or by heterologous expression, it provides information of the limits of substrate recognition by GTRs, and it identifies fluorescent substrates that may be useful to study transport in the vasculature in plants.

A brief summary of the transport mechanism (H+-coupled symporter) and cellular localization of the GTRs in plants should be added to the introduction. I think GTRs are on the plasma membrane and therefore would function in cellular uptake of gluocosinolates.

Overall, the paper is well-written and the results are clear.

Minor points:

The authors suggest that Km values were not determined because there was insufficient fluorescent glucosinolate. Actually, it would be very challenging to use the fluorescent substrates to determine Km. Initial rates would need to be used – samples taken while the transport rate is linear. And quenching may be a problem at higher substrate concentrations. It would be useful to alert the reader that this assay indicates whether the fluorescent substrate is transported but the assay is not quantitative.

In the methods, line 771-782, there are five instances where the work “kulori” is used when the authors mean to say “buffer”.

The second to last sentence in the discussion “similar affinity as natural GSLs” is an over-interpretation of the results. I think it is fine to say that the synthetic GSLs inhibit transport of the natural GSLs indicating a common binding site.

Author Response

Thanks for the kind words on our work.

The first comment from reviewer was regarding some more intro on the GTRs: “A brief summary of the transport mechanism (H+-coupled symporter) and cellular localization of the GTRs in plants should be added to the introduction. I think GTRs are on the plasma membrane and therefore would function in cellular uptake of gluocosinolates.” We have added a paragraph on the transport mechanism and localization of the GTRs in the introduction as suggested, line 54-60:

The GTRs are located in the plasma membrane where they import GSL into the cytosol, and have been suggested to be involved in phloem loading, leaf distribution and rhizosecretion of GSL [7,14–16]. They belong to the Nitrate and Peptide transporter Family (NPF), where most characterized members are proton-coupled symporters with an electrogen transport process [7,17–20]. It has been shown that the GTRs co-transport protons together with glucosinolates, which elicits a net influx of positive charge that can be measured by two-electrode voltage-clamp (TEVC) electrophysiology.

On the minor points, the reviewer commented that: “The authors suggest that Km values were not determined because there was insufficient fluorescent glucosinolate. Actually, it would be very challenging to use the fluorescent substrates to determine Km. Initial rates would need to be used – samples taken while the transport rate is linear. And quenching may be a problem at higher substrate concentrations. It would be useful to alert the reader that this assay indicates whether the fluorescent substrate is transported but the assay is not quantitative.” We have alerted the reader of the challenges in determining Km with fluorescence intensity and/or LCMS quantification. Line 202-206:

However, the limited amount of F-GSL available did not permit such investigations using electrophysiological measurements, which require high amounts of substrate. Additionally, although possible, Km estimation using LCMS or fluorescence based measurements remain imprecise as both represent cumulative assays where one can only approximate initial transport rates.

The reviewer commented on the end of the discussion: “The second to last sentence in the discussion “similar affinity as natural GSLs” is an over-interpretation of the results. I think it is fine to say that the synthetic GSLs inhibit transport of the natural GSLs indicating a common binding site.” The discussion have been updated as suggested, line 325. This end part of the discussion has been moved to a separate conclusion part based on comments from Reviewer 2. The line now states:

We demonstrate that the three F-GSLs, namely GSL-B-NBD, GSL-B-DNS and GSL-A-Dansylamide, are transported actively by GTR1 and GTR2 and provide indications that they compete for a common binding site.

The reviewer commented on a small mistake on words in the methods: In the methods, line 771-782, there are five instances where the work “kulori” is used when the authors mean to say “buffer”.

Methods have been changed to buffer at the suggested lines, now line 821- 827

Besides, we have added a few extra figure references and made small additions to the text to improve the reading (Line 234, 236, 267-268, 311). All changes are made with track changes.  

Reviewer 2 Report

I would like the Conclusions part to be a separate paragraph at the end of the article and to be presented succinctly and clearly.

Author Response

The reviewer suggested the conclusion to be a separate paragraph instead of being part of the discussion. "I would like the Conclusions part to be a separate paragraph at the end of the article and to be presented succinctly and clearly." We have separated our conclusion in a separate paragraph in the end of the article, line 321-328

Besides, manuscript has been amended according to comments by reviewer 1. Additionally very minor changes have been made to improve the reading. All changes are made with track changes.

Author Response

We could not see any comments in the document uploaded by Reviewer.

Manuscript has been amended according to comments by reviewer 1 and 2. Additionally very minor changes have been made to improve the reading. All changes are made with track changes.

Round 2

Reviewer 2 Report

I accept in this form.